# Expression of Autoimmunity-Related Genes in Melanoma

**DOI:** 10.3390/cancers14040991

**Published:** 2022-02-16

**Authors:** Francesca Scatozza, Antonio Facchiano

**Affiliations:** Laboratory of Molecular Oncology, Istituto Dermopatico dell’Immacolata, Istituto di Ricovero e Cura a Carattere Scientifico (IDI-IRCCS), Via Monti di Creta 104, 00167 Rome, Italy; f.scatozza@idi.it

**Keywords:** autoimmunity, melanoma, *NOD2*, *BAX*, *IL-18*, *ADRB2*

## Abstract

**Simple Summary:**

The present study selected four genes strongly related to autoimmunity. Their expression was found to be significantly altered in melanoma patients according to a multi-validation procedure carried out on 1948 patients. Such genes may represent suitable molecular targets to further investigate the role autoimmunity may play in melanoma setup and development. Our data suggest that autoimmunity may play a beneficial role in melanoma set up, at least to some extent.

**Abstract:**

(1) Background. Immune response dysregulation plays a key role in melanoma, as suggested by the substantial prognosis improvement observed under immune-modulation therapy. Similarly, the role of autoimmunity is under large investigation in melanoma and other cancers. (2) Methods. Expression of 98 autoimmunity-related genes was investigated in 1948 individuals (1024 melanoma and 924 healthy controls). Data were derived from four independent databases, namely, GEO in the selection phase, and Ist Online, GEPIA2 and GENT2, in three sequential validation-steps. ROC analyses were performed to measure the ability to discriminate melanoma from controls. Principal Component Analysis (PCA) was used to combine expression data; survival analysis was carried out on the GEPIA2 platform. (3) Results. Expression levels of *NOD2*, *BAX*, *IL-18* and *ADRB2* were found to be significantly different in melanoma vs. controls and discriminate melanoma from controls in an extremely effective way, either as single molecules (AUC > 0.93 in all cases) or as a profile, according to the PCA analysis. Patients showing high-expression of *NOD2* and of *IL-18* also show a significant survival improvement as compared to low-expression patients. (4) Conclusions. Four genes strongly related to autoimmunity show a significant altered expression in melanoma samples, highlighting the role they may play in melanoma.

## 1. Introduction

Autoimmunity is activated when the immune system recognizes target cells as non-self. Under normal conditions and activity, the immune system reacts towards foreign molecules that are potentially harmful; for example, against molecules associated with viruses, bacteria and parasites. Under such conditions, structures recognized as self undergo immune tolerance [1]. On the other hand, if autologous cells or cellular components are recognized as non-self, the production of autoantibodies occurs, eventually leading toward autoimmune diseases. Autoantibodies may be directed against a single cell type or tissue type, and such autoantibodies cause organ-specific diseases such as Addison’s disease, celiac disease, Graves disease, Hashimoto thyroiditis and type I diabetes mellitus. Alternatively, autoantibodies may be directed against components common to all cells, causing systemic diseases, such as systemic lupus erythematosus or rheumatoid arthritis. The presence of autoantibodies has been demonstrated in patients with psoriasis, even in clinically silent phases [2], in essential hypertension [3] and pulmonary hypertension [4]. The possible connection between cancer and autoimmune disorders has been investigated for several years [5]. Although an autoimmune pathogenesis is not demonstrated in melanoma, a relation between the immune response to melanoma and the presence of autoimmune reactions is reported in different studies [6,7,8,9]. A study carried out on 20,482 cases of metastatic cutaneous melanoma showed that patients with a previous diagnosis of autoimmune disease (e.g., diabetes) have a 30% higher incidence of metastatic cutaneous melanoma in the first year of diagnosis, as compared to individuals without autoimmune disease [10]. Further, it is known that patients with Retinopathy Associated with Melanoma (MAR) have a high prevalence of autoimmune diseases in their families [11]. Furthermore, in melanoma patients undergoing therapy with interferon alpha, the appearance of autoantibodies is associated with a better prognosis [12,13], demonstrating that an autoimmune reaction is not neutral in melanoma progression. Although conflicting data are present in the literature on this aspect [14], such data demonstrate that melanoma onset and/or development as well as response to melanoma therapy has several relation points with autoimmunity. While specific mutations in autoimmunity-related genes are reported in melanoma, [15,16], gene expression levels have not been deeply investigated, under such respect. We therefore investigated the expression of autoimmunity related genes in melanoma patients as compared to healthy controls. To this aim the expression of 98 genes was investigated in 1948 individuals (1024 melanoma patients and 924 healthy controls), derived from four independent datasets, namely GDS1375 dataset from GEO database, IST Online, GEPIA2 and GENT2.

## 2. Results

### 2.1. The General Scheme of the Analysis Performed in The Present Study Is Reported

Figure 1 synthetizes the followed procedure.

### 2.2. Knowledge-Based Autoimmunity Score (kb-AS) in Different Tumors

Melanoma is reported as being strongly related to autoimmune responses. In PubMed-indexed manuscripts we measured the occurrence of autoimmunity keywords in manuscripts related to 28 different cancer types, according to the methodology reported in Materials and Methods. Such occurrence is here reported as a “Knowledge-based Autoimmunity Score” (kb-AS). Table 1 reports the 28 cancer types sorted from the lowest to the highest kb-AS. This analysis indicates cancer types known to be mostly related to autoimmunity and confirmed that melanoma is in the top five group, along with Thymoma, Lymphoma, Thyroid cancer and Cholangiocarcinoma.

### 2.3. Expression of Autoimmunity-Related Genes in GEO Database, in Melanoma vs. Controls

Ninety-eight autoimmunity related-genes were selected from the literature [17,18,19] and their expression levels in melanoma vs. control samples were analyzed in the GDS1375 dataset in the GEO database (https://www.ncbi.nlm.nih.gov/sites/GDSbrowser?acc=GDS1375) accession date 8 April 2021. Table 2 reports the mean values in melanoma and in control samples. The fold change in melanoma vs. controls is also reported when it is <0.5 or >1.5. When this requirement is matched, the AUC (Area Under the Curve) is reported if >0.8, according to the ROC (Receiver Operating Characteristic) analysis; *p* value is also indicated. AUC indicates the ability to discriminate melanoma from nevi samples; its value ranges from 0.5 to 1, with 1 indicating 100% ability to discriminate controls from disease cases.

Severe requirements were selected from the values reported in Table 2 to identify differential expression in melanoma vs. controls able to effectively differentiate melanoma from control samples. Namely, the requirements were: fold change <0.5 or >1.5; AUC > 0.93, *p* < 0.0001. Six genes match these criteria, namely: *NOD2*, *BAX*, *IL-18*, *ADRB2*, *ITGAV* and *MYO9B*. The ROC analysis for each gene is reported in Figure 2, with the indication of AUC and *p* values. The AUC is > 0.93 in all cases, indicating an extremely high ability to differentiate melanoma samples from controls.

### 2.4. First Validation: Gene Expression Analysis in IST Online Database

The expression of the six genes (*NOD2*, *BAX*, *IL-18*, *ADRB2*, *ITGAV* and *MYO9B*) was then investigated in an independent public database, namely IST Online. Such database contains several different cancers datasets and normal tissues datasets; the melanoma dataset contains 208 melanoma patients and the healthy skin dataset, used here as control, contains 147 healthy skin samples. Data collected on the GEO dataset were fully validated; in fact, all six genes confirmed significant different expression levels in melanoma vs. healthy skin, in the IST Online dataset (Figure 3), according to the Fisher’s exact test analysis (*p* < 0.0001). The used threshold values are indicated as dashed lines in Figure 3. We then concluded that all genes show a strongly different expression in melanoma vs. control biopsies, in both GDS1375 and IST Online datasets.

### 2.5. Second Validation: Gene Expression Analysis in GEPIA2 Database

During the finalization of the present study, Ist Online database became not publicly available any longer. We then decided to validate the six genes in two additional public databases. As a second validation, the expression levels of *NOD2*, *BAX*, *IL-18*, *ADRB2*, *ITGAV* and *MYO9B* were analyzed in GEPIA2 database (available at http://gepia2.cancer-pku.cn/#index, accessed on 17 May 2021), a public database reporting expression data from 461 melanoma patients and 558 healthy skin controls. Four out of six genes showed a significantly different expression in melanoma vs. healthy skin, as depicted in Figure 4.

### 2.6. Third Round Validation on GENT2 Database

As a third-round validation, the expression levels of the four genes *NOD2*, *BAX*, *IL-18* and *ADRB2* were investigated in skin cancer samples from the GENT2 database (available at http://gent2.appex.kr/gent2/ accession on 28 May 2021), which collects data from 310 skin cancers samples and 201 controls. All four genes show a strongly significant differential expression in skin cancers as compared to the appropriate controls. Namely *NOD2* shows a LOG2 FC (Fold change expressed in Log2) expression of −1.34, *p* < 0.001; *BAX* shows a LOG2FC expression of 1.3, *p* < 0.001; *IL-18* shows a LOG2FC expression of 3.09, *p* < 0.001; *ADRB2* shows a LOG2FC expression of 1.6, *p* < 0.001.

### 2.7. Principal Component Analysis (PCA) and Multiple Logistic Analysis

We have previously used Principal Component Analysis (PCA) to identify the best genes-combination able to discriminate melanoma from controls within a large selection of nicotinamide receptors [20] and cytokines/chemokines [21]. The combined expression level of the four validated genes (*NOD2*, *BAX*, *IL-18* and *ADRB2*) was then analyzed according to the PCA dimensionality reduction tool in the GEPIA2 database. As reported in Figure 5, the three-dimensional space defined by the three variance components (PC1, PC2 and PC3) associated with the expression values of these four genes show a striking separation between 461 melanoma and 558 healthy controls, indicating that the expression profile of such four genes strongly differentiates melanoma from healthy controls.

As an additional evaluation, multiple logistic analysis and ROC analysis were carried out on the four genes combined expression profile, in the 45 melanoma and 18 nevi samples derived from the GEO GDS1375 dataset. Figure 6 reports the ROC curve and demonstrates that combining the expression values of *NOD2*, *BAX*, *IL-18* and *ADRB2* achieves an extremely high efficacy to discriminate controls from melanoma samples, with AUC = 0.983. In addition, 88.9% of observed nevi were correctly classified and 93.3% of observed melanoma were correctly classified.

### 2.8. NOD2, BAX, IL-18 and ADRB2 Gene Expression in Different Cancer Types

We then further characterized the gene-expression of the four validated genes in 28 different cancer types. Table 3 reports the tumors sorted from the lowest to the highest “Knowledge-based Autoimmunity Score” (kb-AS) according to Table 1. As indicated by the asterisks, expression of *NOD2*, *BAX*, *IL-18* and *ADRB2* is significantly altered mostly in cancer types showing the highest autoimmunity scores, further supporting the possible role of such genes in the interplay between autoimmunity and cancer.

### 2.9. Survival in Patients Expressing High- and Low- Levels of NOD2, BAX, IL-18 and ADRB2

The survival associated with the expression values of each gene was then analyzed according to the GEPIA2 database. Patients expressing high levels of *NOD2* or *IL-18* showed a significantly better survival, with a Hazard Ratio (HR) of 0.58 and 0.52, respectively, compared to low expression patients (Figure 7); the analysis was performed on 229 high-expressing patients and 229 low-expressing patients. On the contrary, expression of *BAX* or *ADRB2* has no significant impact on patients’ survival (data not shown). In addition to the possible diagnostic role reported in the previous figures, such data indicate that NOD2 and IL-18 expression levels may also have a prognostic value in melanoma.

## 3. Discussion

A possible role of autoimmunity in melanoma is currently under intense investigation. Autoimmunity is generally recognized as a possible severe side effect of melanoma therapy, upon interferon- [22] or ipilimumab-treatment [23]. On the other hand, unchaining the immune system is the main goal of current immunotherapy protocols as well as previous approaches combining dacarbazine with IL-2 and GM-CSF [24]. Data published in 2018 indicate a high prevalence rate of pre-existing autoimmune comorbidities in melanoma patients as compared to the general population (28.3% and 19.8% in metastatic and not-metastatic melanoma, respectively, as compared to 9.2% in the general population) [25]. Other studies published in 2013 show that the co-occurrence of melanoma with other diseases, including autoimmune diseases, increases the melanoma-mortality rate [26], indicating a possible etiopathogenetic role of autoimmune response in the melanoma onset. Nevertheless, a significant prevalence of autoimmune diseases in melanoma was questioned in a previous study published in 2007 [10]. It is commonly reported that a dysregulation of the immune response, including autoimmune conditions, may predispose toward the cancer development, likely due to the chronic inflammation occurring in these patients [8].

On the other hand, autoimmunity may also have a protective role. In fact, patients undergoing an autoimmune reaction against melanocytes, such as vitiligo patients, show either a lower risk of developing melanoma [27,28] or a better prognosis [29,30], most likely due to a so-called “beneficial autoimmunity” [31]. The present study investigated the hypothesis that the expression of genes directly involved in autoimmunity may be altered in melanoma patients and may therefore represent suitable molecular targets for possible diagnostic, prognostic or therapeutic applications. To address this hypothesis, a non-exhaustive list of 98 genes related to autoimmunity [17,18,19] was investigated while the proper bioinformatic tools are being developed to address the complete list of 4186 genes contained in the GAAD autoimmunity-related genes database [19]. A sequential multi-step validation was carried out on four independent public databases (GEO, IST Online, GEPIA2 and GENT2) involving a total of 1024 melanoma patients and 924 controls. Four genes were identified and validated in all databases, showing a significantly altered expression in melanoma vs. controls, namely *NOD2*, *BAX*, *IL-18* and *ADRB2*, *NOD2* and *IL-18* appear to be particularly interesting from a prognostic point of view since their expression was also found to be related to a significant survival improvement (Figure 7). *NOD2* is a member of the Nod1/Apaf-1 family; the encoded protein has two caspase recruitment domains and six leucine-rich repeats giving to NOD2 a protein-binding activity. Leucin-rich repeats are functionally related to innate immunity activity, functioning as sensors of pathogen-associated molecular patterns [32] and triggering NF-kB activation upon recognition of bacterial lipopolysaccharides. *NOD2* haplotypes have been associated with Crohn disease [33], and a gain-of-function mutation of *NOD2* occurs in patients with Blau syndrome, a rare systemic granulomatous inflammatory disease [34]. Both Chron disease and Blau syndrome are reported to have a clear autoimmunity basis [35,36]. Several polymorphisms of *NOD2* are known; some are recognized to be related to increased cancer risk, nevertheless melanoma appears to not be significantly related to NOD2 polymorphisms [37,38]. The present study suggests for the first time that *NOD2* expression levels may be indeed related to melanoma onset. Interestingly, patients with Chron disease have an increased risk of developing melanoma and other cancers [39] and at least two cases of Blau Syndrome in melanoma patients have been described [40]. In all four databases investigated, *NOD2* expression appears strongly and significantly reduced in melanoma patients, suggesting that the loss of its beneficial function in the immune and autoimmune surveillance may have a role in melanoma. Its protective role may be further recognized since high-expression patients show better survival, as compared to low-expression patients (Figure 7). *NOD2* is involved in many processes such as embryogenesis [41] and in regeneration [42] and chronic inflammation [43], suggesting that the relationship between *NOD2* and melanoma/cancer may not be limited to the immune system, but may involve also other mechanisms related to carcinogenesis and the evolution of malignancy [44].

*IL-18* is one of the factors most related to autoimmunity [45,46,47,48] due to its potent and complex pro-inflammatory action. *IL-18* is also known to play a key role in melanoma regulation [49] and is one of the AIM2 inflammasome components [50], recently recognized as a potential target for melanoma treatment [51]. Inducing *IL-18* expression is known to exert an anti-melanoma effect in mice [52]; this is compatible with our data showing a significant reduction of *IL-18* expression in melanoma patients, according to all four databases investigated, and is consistent with the better survival observed in patients showing high *IL-18* expression (Figure 7). The role of *IL-18* expression in melanoma patients has been previously reported [53], but the interplay with the autoimmunity is highlighted here for the first time, in the melanoma setting.

*ADRB2* codes for the adrenoreceptor beta 2 G protein-coupled receptor. It has a central role in controlling immune and autoimmune processes, and in suppressing autoimmunity in the central nervous system [54]. As recently underlined [55], *ADRB2* plays a dual role in inducing or inhibiting autoimmune disorders at both the systemic and local level, exerting opposite actions in different stages of autoimmune diseases such as Rheumatoid Arthritis [55]. Polymorphisms of this gene are related to susceptibility to autoimmune diseases such as Graves disease [56] and Rheumatoid Arthritis [57]. *ADRB2* has been previously found associated with melanoma [58]; the present study investigates its expression in a melanoma onset, for the first time. The strong and significant reduced expression in melanoma patients suggests a loss-of-function and a reduced immune or autoimmune control toward melanoma-related antigens. *ADRB2* is closely related to the C L type calcium channel Ca (V)1.2, further supporting the role we recently underlined ion channels play, as possible melanoma therapeutic targets [59,60].

*BAX* is a proapoptotic molecule, member of the BCL-2 family, with a recognized role to control immune tolerance and prevent autoimmune disorders in mice [61]. Its complete deletion in MCL-1 depleted cells has recently been found to completely rescue a lethal multiorgan autoimmunity [62], supporting a key role in the control of immune tolerance. We report here a significantly increased expression of *BAX* in melanoma patients, supporting it as a possible autoimmunity-related target in melanoma.

The link between autoimmunity and cancer is largely investigated but it is still controversial whether autoimmunity helps or blocks cancer set up or development [31,63,64]. Further investigations are needed to clarify also the potential therapeutic applications of drugs targeting autoimmunity.

Autoimmune manifestations occur in melanoma, which is known to express self-antigens such as B melanoma antigen 1 (BAGE) and melanoma-associated antigens (MAGEs). The co-occurrence of vitiligo, an autoimmunity-based disease, has been associated with better prognosis in melanoma patients [65] and a preexisting vitiligo is associated with a significantly reduced incidence of melanoma [66] and of other cancers [67]. Furthermore, vitiligo regression associates with melanoma progression [68]. All such data strongly support the hypothesis that autoimmunity may be beneficial for melanoma, as well as other cancer types, as reviewed by a recent study by Zitvogel et al. [31]. Data reported in the present study support the hypothesis of a beneficial autoimmunity mediated by *NOD2*, *IL-18* and *ADRB2*, since their expression is shown to be almost abolished in melanoma patients. Interestingly, the multiple logistic analysis shows that the AUC = 0.98 computed for the four-genes combined profile (Figure 6), is much higher than the AUC computed for the single genes (AUC = 0.95 for *NOD2*, *IL-18* and *ADRB2*; AUC = 0.93 for *BAX*) (Figure 2). This demonstrates that combining the expression values is much more effective to discriminate controls from melanoma, as compared to the values of the single genes.

Finally, when the expression analysis was extended to several cancer types other than melanoma, the four genes were found significantly altered especially in cancer types mostly related to autoimmunity, according to the knowledge-based Autoimmunity Score (kb-AS) computed in the present study. Some pitfalls may be recognized in the procedure followed to measure kb-AS. The first relates to the continuous growth of the PubMed database, which implies the possibility that the kb-AS may (or will) change over time. Further, less common cancer types were not investigated. We are aware that the searches carried out were not extensive, and we will address this issue in more details and larger completeness in a study focused on this specific aspect. However, we believe the current form of the kb-AS is sufficiently indicative to recognize at least the most common cancer-types showing the known highest relation to autoimmunity.

## 4. Materials and Methods

Ninety-Eight Autoimmunity-Related Genes Were Taken for the Literature [17,18,19]. The genes investigated are known to be related to allergies, ankylosing spondylitis, asthma celiac diseases, Crohn’s disease, type 1 and type 2 diabetes, Graves’ diseases, inflammatory bowel disease, multiple sclerosis, psoriasis, rheumatoid arthritis, Sjögren syndrome, systemic lupus erythematosus and ulcerative colitis. The expression of such genes was investigated in a total of 1948 human samples derived from four different public datasets, via one initial screening phase and three additional validation phases.

### 4.1. Calculating the “Knowledge-Based Autoimmunity Score” kb-AS

The known relation of 28 cancer types to autoimmunity was measured as follows: PubMed searches were carried out looking for the occurrence of the words relating to each cancer type in All-fields. Such a procedure was carried out as a stand-alone search or in the presence of autoimmunity-related words. Therefore, the co-occurrence of cancer-related words and autoimmunity-related words was measured. Words used to identify each cancer type and to identify autoimmunity are reported below.

The number of manuscripts presenting the co-occurrence, divided by the total number of manuscripts related to each cancer-type, expressed as percentage, was then reported as the “knowledge-based Autoimmunity Score” of cancers (kb-AS). The cancer types investigated were the 28 cancer types reported in the GEPIA2 database. The words related to each cancer type, searched in PubMed All-fields, are as follows:

Cervical = “Cervical squamous cell carcinoma” OR “Endocervical adenocarcinoma”;

Esophag = “Esophageal carcinoma” OR “Esophagus carcinoma”;

Pheoc-Para = “Pheochromocytoma” OR “Paraganglioma”;

Endometrial = “Uterine Corpus Endometrial carcinoma” OR “endometrial carcinoma” OR “endometrial cancer” OR “Endometrium cancer” OR “Uterine carcinosarcoma”;

Rectum = “Rectum adenocarcinoma” OR “rectum carcinoma” OR “rectum cancer”;

Adrenocortical = “Adrenocortical carcinoma”;

Prostate = “Prostate adenocarcinoma” OR “prostate carcinoma” OR “prostate cancer”;

Mesothelioma = “Mesothelioma”;

Ovary = “Ovarian serous cystadenocarcinoma” OR “Ovarian carcinoma” OR “Ovarian adenocarcinoma”;

Head and Neck = “Head and Neck squamous cell carcinoma” OR “Head and Neck carcinoma”;

Gliobl = “Glioblastoma”;

Sarcoma = “Sarcoma”;

Liver = “Liver hepatocellular carcinoma” OR “liver carcinoma”;

Bladder = “Bladder carcinoma”;

Testis = “testicular Germ cell tumors” OR “testicular cancer” OR “testis cancer “ OR “seminoma”; Colon = “Colon adenocarcinoma”;

Kidney = “Kidney carcinoma” OR “kidney cancer”;

Breast = “Breast carcinoma”;

Lung = “Lung adenocarcinoma” OR “Lung carcinoma”;

Glioma = “Glioma”;

Leukemia = “Leukemia”;

Gastric = “Gastric adenocarcinoma” OR “stomach adenocarcinoma”;

Pancreati c = “Pancreatic adenocarcinoma” OR “pancreatic carcinoma”;

Melanoma = “Melanoma”;

Chol = “Cholangio carcinoma”;

Thyroid = “Thyroid carcinoma”;

Lymphoma = “Lymphoma”;

Thymoma = “Thymoma”.

The autoimmunity related words with wild cards indicated by the asterisks were: “autoimmun*” OR “autoantigen*” OR “selfantigen*” OR “selftolerance”.

### 4.2. Expression of Autoimmunity-Related Genes, in GEO Database: Melanoma vs. Nevi Samples

The expression of 98 autoimmunity-related genes was evaluated in the melanoma GDS1375 dataset, from the GEO public database (https://www.ncbi.nlm.nih.gov/gds/), accessed on 8 April 2021 reporting the actual expression values in 63 samples (45 melanoma-patients vs. 18 nevi-patients). Calculations such as mean, Mann–Whitney test and ROC analysis were then carried out. ROC analysis is a binary test; it was used here to measure how effective the expression-level of any given gene to discriminate healthy from melanoma-biopsies is. The computed Area Under Curve (AUC) value ranges from 0.5 to 1, indicating 50% to 100% discrimination ability.

### 4.3. First Validation of Gene Expression Data in IST Online Database

The first validation step was carried out by analyzing expression levels of the selected genes in the independent IST Online database (http://ist.medisapiens.com/) accessed on 6 May 2021, having 208 melanoma samples and 147 healthy-skin samples. Different to the GEO database, IST Online does not show actual numbers; rather it expresses data as scatter plots, where each dot is one patient. Genes showing expression levels in melanoma significantly different to controls, according to the Fisher’s exact test, were then considered validated and were selected for the following validation step.

### 4.4. Second Validation of Gene Expression Data in GEPIA2 Database

During the finalization of the present study, the IST Online database, previously public for many years, became unavailable. In order to assure a full reproducibility of the results, we therefore decided to carry out additional validation steps. A second round validation step was carried out on the public database GEPIA2, available at http://gepia2.cancer-pku.cn/#index [69] accessed on 17 May 2021, having 461 melanoma and 558 healthy-skin samples. Genes matching stringent significance threshold parameters (namely, Log2 [FC] > 1.5 and *p* value < 0.0001) were considered fully validated.

### 4.5. Third Validation of Gene Expression Data in GENT2 Database

The four selected genes (*NOD2*, *BAX*, *IL-18* and *ADRB2*) were investigated in control and in skin cancer samples (total 1042 samples, including 201 controls and 310 melanoma samples), taken from the GENT2 database at the link http://gent2.appex.kr/gent2/ accessed on 28 May 2021. The gene profile tools were used, indicating as types: “Tissue”, terms: “Gene symbol”, and as keyword the four genes symbol; data were then extracted from the table reporting T-test and Log2 fold changes results.

### 4.6. PCA Analysis and Multiple Logistic Analysis

The “dimensionality reduction” tool available on GEPIA2 at http://gepia2.cancer-pku.cn/#dimension accessed on 10 June 2021, was exploited to perform the PCA analysis [70] combining *NOD2*, *BAX*, *IL-18* and *ADRB2* expression levels, according to the methodology previously used [19,20]. Log scale was used; cancer type was SKCM tumor (Skin Cutaneous Melanoma); controls were SKCM normal, skin not sun exposed, skin sun exposed.

Multiple logistic analysis was carried out using the expression values of *NOD2*, *BAX*, *IL-18* and *ADRB2* in controls and melanoma patients, and ROC analysis on their combined expression values was computed.

### 4.7. Survival Analysis in Melanoma Patients

Survival analysis in melanoma patients was investigated according to GEPIA2 at http://gepia2.cancer-pku.cn/#survival accessed on 11 June 2021. The analysis was performed on 229 high-expressing patients and 229 low-expressing patients. Survival analyses were carried out on quartile distribution with the cutoff for high expression set at 70% and the cutoff for low expression set at 25%, using the skin cancer melanoma (SKCM) dataset, according to a methodology previously described [20].

### 4.8. Expression of NOD2, BAX, IL-18 and ADRB2 in 28 Cancer Types

The expression levels of *NOD2*, *BAX*, *IL-18* and *ADRB2* were analyzed in all cancer types present in GEPIA2 database, available at http://gepia2.cancer-pku.cn/#index [69] accessed on 1 July 2021. Log2 [FC] > 1.5 and *p* value < 0.0001 were selected as the significance threshold.

### 4.9. Statistics

The analyses were performed on 1948 human samples from four independent databases. The samples distribution was as follows: 45 melanoma and 18 nevi samples from GEO database; 208 melanoma and 147 healthy-skin samples from IST Online database; 461 melanoma and 558 healthy-skin samples from GEPIA2 database and 310 melanoma and 201 healthy skin samples from GENT2 database. Mean, two tails t Test, Mann–Whitney test and ROC analysis were carried out, with a significance-threshold set at *p* value < 0.0001. Fisher’s exact test was carried out to evaluate sample distribution above or below a given threshold, in data from Ist Online database. The statistical package GraphPad Prism version 9.3.0 was used (Graph Pad software Inc, 2365 Northside Dr., Suite 560, San Diego, CA 92108, USA).

## 5. Conclusions

In conclusion, the present study identifies four genes strongly related to autoimmunity. Their expression was found to be significantly altered in melanoma patients according to a multi-validation procedure carried out on 1948 patients from four independent databases. We believe such genes may represent suitable molecular targets to further investigate the role autoimmunity may play in melanoma setup and development. Such data suggest that autoimmunity may play a beneficial role in melanoma set up, at least to some extent.

## Figures and Tables

**Figure 1 cancers-14-00991-f001:**
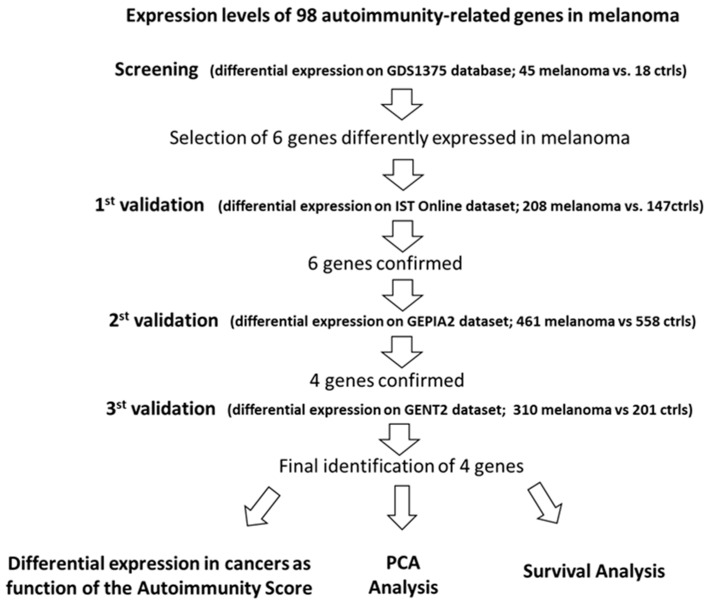
Summary for the selection of the best autoimmunity-related candidate-genes in melanoma.

**Figure 2 cancers-14-00991-f002:**
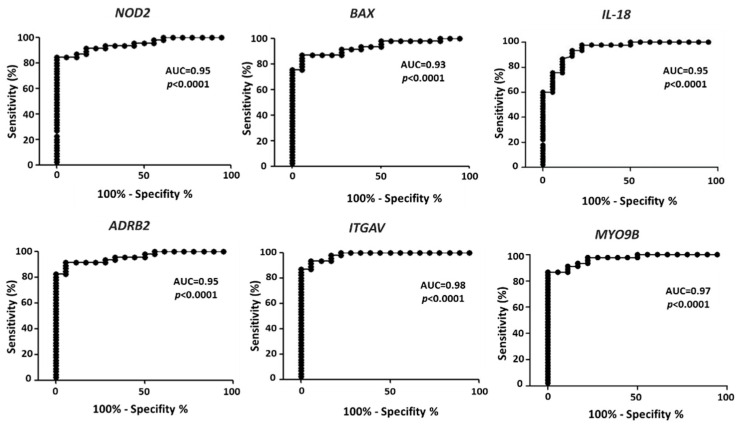
ROC analysis on the expression values of the six selected genes, according to the expression values reported in GDS1375 from GEO Database. The area under the curve (AUC) is plotted as sensitivity% vs. 100-specificity%. The calculated AUC and *p* value are reported in each case.

**Figure 3 cancers-14-00991-f003:**
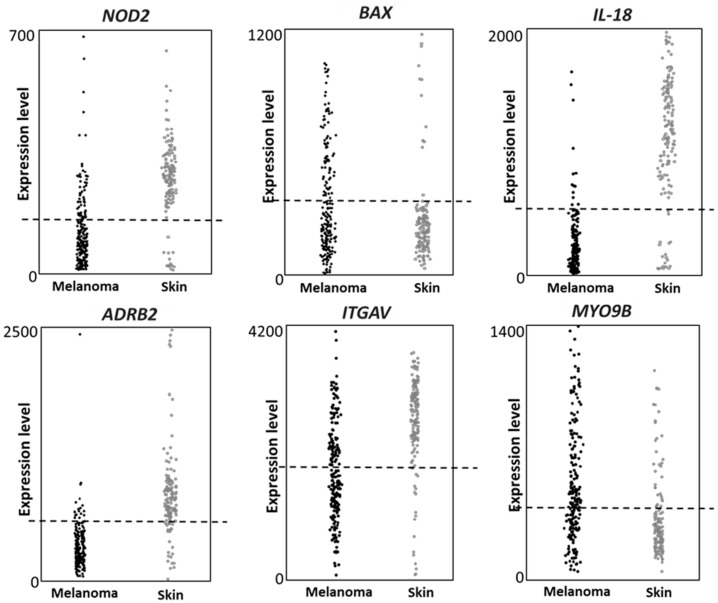
Gene expression according to the IST online database. The six genes show different expression levels in melanoma vs. healthy skin. The expression level of each gene is reported in 208 melanoma biopsies and 147 healthy skin biopsies. The Fisher analysis shows for each gene a *p* value < 0.0001. The expression value threshold was: 200 for *NOD2* and *BAX*; 500 for *ADRB2*, *IL-18* and *MYO9B*; 2000 for *ITGAV*.

**Figure 4 cancers-14-00991-f004:**
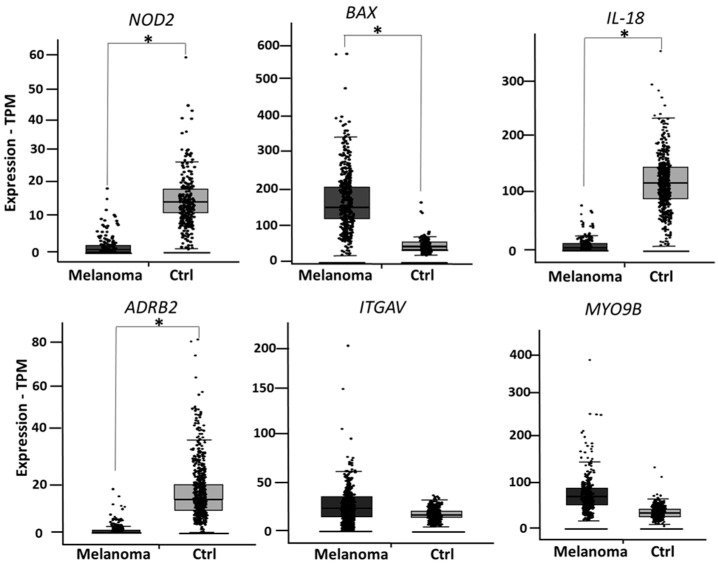
According to GEPIA2 database, *NOD2*, *BAX*, *IL-18* and *ADRB2* show a significantly different expression in melanoma vs. healthy skin, while expression levels of *ITGAV* and *MYOB9* are not significantly changed. * indicates *p* < 0.0001.

**Figure 5 cancers-14-00991-f005:**
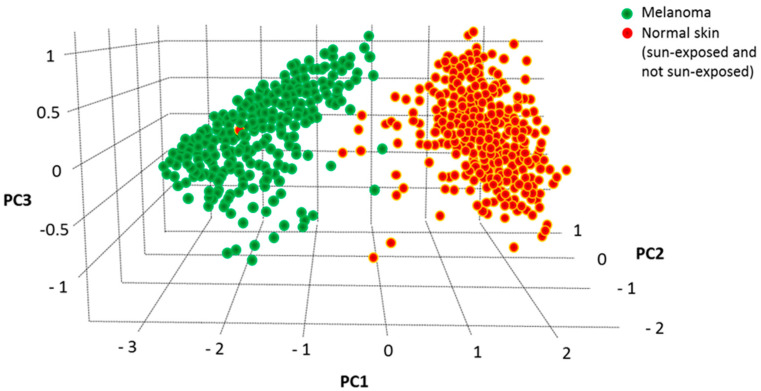
PCA analysis of the combined values of *NOD2*, *BAX*, *IL-18* and *ADRB2* carried out according to GEPIA2 (http://gepia2.cancer-pku.cn/#dimension accession at 10 June 2021). Melanoma samples appear clearly separated from controls in the three-dimensional space, indicating that the combined expression values of these four genes shows distinct variance in melanoma vs. healthy controls.

**Figure 6 cancers-14-00991-f006:**
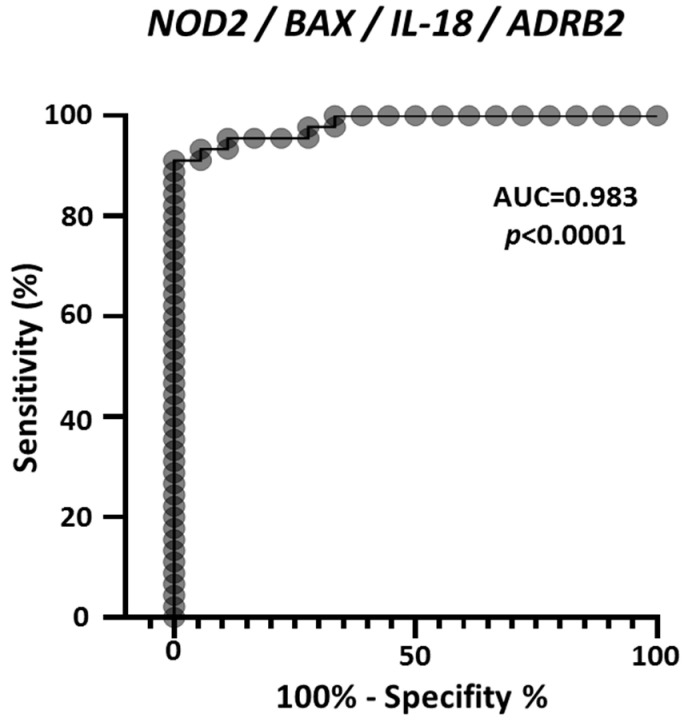
ROC analysis on the combined expression values of *NOD2*, *BAX*, *IL-18* and *ADRB2*, according to the values reported in GDS1375 from the GEO Database. The area under the curve (AUC) is plotted as sensitivity% vs. 100%-specificity%. The calculated AUC and *p* value are reported.

**Figure 7 cancers-14-00991-f007:**
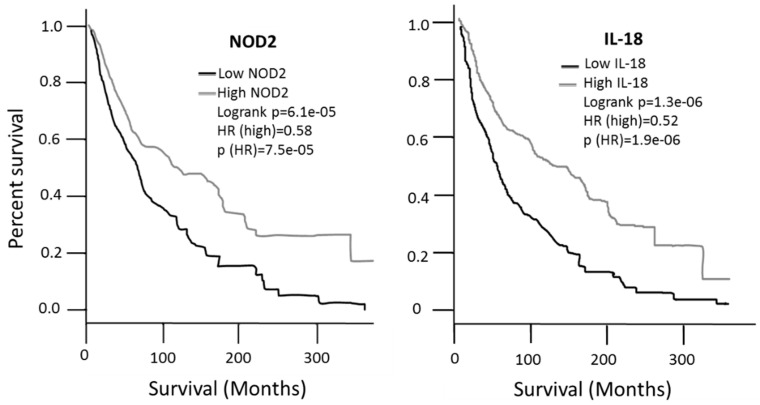
Survival analysis carried out in the GEPIA2 database. *NOD2* and *IL-18* expression values show a significantly different survival in patients with high vs. low expression patients.

**Table 1 cancers-14-00991-t001:** The “Knowledge-based Autoimmunity Score” (kb-AS) was calculated from the PubMed occurrence of autoimmunity-related words within cancer manuscripts. The kb-AS (column C) is expressed as a percentage of the co-occurrence (column B) on the total number of cancer-related manuscripts (column A). Cancer types were those present as cancer-datasets in the GEPIA2 database. The full names of each cancer type are reported in Material and Methods section. The occurrence is reported as number of manuscripts in PubMed, according to the search carried out on 15th November 2021 in the ALL-FIELDS field.

		A	B	C
Rank	Cancer Type	Occurrence of Cancer Type Alone *	Co-Occurrence with Autoimmunity Keywords in Cancer Types *	Knowledge-Based Autoimmunity Score (kb-AS) (%)
28th	Cervical	1806	1	0.055
27th	Esophagus	6433	8	0.124
26th	Pheoc-Para	29,317	43	0.147
25th	Endometrial	25,760	55	0.213
24th	Rectum	1156	3	0.260
23th	Adrenocortical	3461	9	0.260
22th	Prostate	134,130	372	0.277
21th	Mesothelioma	19,470	61	0.313
20th	Head&Neck	12,287	45	0.366
19th	Ovary	14,914	56	0.370
18th	Glioblastoma	46,066	178	0.386
17th	Liver	1983	8	0.403
16th	Sarcoma	120,133	488	0.406
15th	Bladder	6397	27	0.422
14th	Testis	13,194	56	0.424
13th	Colon	4930	22	0.446
12th	Kidney	6484	29	0.447
11th	Breast	26,055	136	0.521
10th	Lung	95,990	593	0.617
9th	Glioma	65,563	469	0.715
8th	Gastric	7927	71	0.896
7th	Leukemia	325,600	4100	1.259
6th	Pancreatic	15297	203	1.327
5th	Melanoma	145,114	2169	1.494
4th	Cholangio	16,805	291	1.731
3th	Thyroid	23,065	431	1.868
2th	Lymphoma	233,219	5657	2.425
1th	Thymoma	11,523	1197	10.38

* number of manuscripts.

**Table 2 cancers-14-00991-t002:** Mean expression values of the 98 autoimmunity-related genes in melanoma and control samples, according to the GDS1375 dataset. The fold change in melanoma vs. controls is reported when it is <0.5 or >1.5. In such cases, the AUC (Area Under the Curve) is indicated, if it is >0.8, according to the ROC (Receiver Operating Characteristic) analysis; also *p* values are reported in such cases (n.s = not significant).

	Symbol	Mean Express. in Melanoma	Mean Express. in Nevi	Fold Change Melan. vs. Nevi	AUC	*p* Value
1	** *ABCB1* **	127	141	-	n.s.	n.s.
2	** *ADRB2* **	**169**	**552**	**0.30**	**0.96**	**<0.0001**
3	** *ALK* **	147	128	-	n.s.	n.s.
4	** *ANKRD1* **	30	43	-	n.s.	n.s.
5	** *ARTS1* **	213	297.3	-	n.s.	n.s.
6	** *ATG16L1* **	84.3	92.7	-	n.s.	n.s.
7	** *AXL* **	345	623	-	n.s.	n.s.
8	** *BANK1* **	46.61	70.2	-	n.s.	n.s.
9	** *BATF* **	425	406	-	n.s.	n.s.
10	** *BAX* **	**498**	**186**	**2.67**	**0.93**	**<0.0001**
11	** *BLK* **	184	190	-	n.s.	n.s.
12	** *BSN* **	49	49	-	n.s.	n.s.
13	** *CBLB* **	608	564	-	n.s.	n.s.
14	** *CCL21* **	**167**	**719**	**0.23**	**0.85**	**<0.0001**
15	** *CD40* **	793	785.36	-	n.s.	n.s.
16	** *CD44* **	**285**	**709**	**0.40**	**0.86**	**<0.0001**
17	** *CD58* **	573	392	-	n.s.	n.s.
18	** *CD81* **	11,099	8238	-	n.s.	n.s.
19	** *CD8A* **	280	357	-	n.s.	n.s.
20	** *CDKAL1* **	176	118	-	n.s.	n.s.
21	** *CDKN2A* **	**1231**	**710**	**1.73**	**0.89**	**<0.0001**
22	** *CDKN2B* **	27	20	-	n.s.	n.s.
23	** *CLMN* **	83	156	-	n.s.	n.s.
24	** *CMT1B* **	181	186	-	n.s.	n.s.
25	** *COL4A3* **	127	150	-	n.s.	n.s.
26	** *CTLA4* **	104	138	-	n.s.	n.s.
27	** *CXCL8* **	586	43	13.48	n.s.	n.s.
28	** *DBC1* **	158	168	-	n.s.	n.s.
29	** *ECM1* **	2843	1320	2.15	n.s.	n.s.
30	** *ERBB3* **	140	179	-	n.s.	n.s.
31	** *EVI5* **	204	276	-	n.s.	n.s.
32	** *FAM69* **	694	536	-	n.s.	n.s.
33	** *FASLG* **	171	180	-	n.s.	n.s.
34	** *FCGR2A* **	395	282	-	n.s.	n.s.
35	** *FCGR3B* **	418	396	-	n.s.	n.s.
36	** *FEN1* **	839	690	-	n.s.	n.s.
37	** *GADD45A* **	574	663	-	n.s.	n.s.
38	** *HHEX* **	176	200	-	n.s.	n.s.
39	** *HIF1A* **	2397	2230	-	n.s.	n.s.
40	** *HLAB* **	11,301	13,761	-	n.s.	n.s.
41	** *HLAC* **	4420	5133	-	n.s.	n.s.
42	** *HLADQA1* **	206	613	-	n.s.	n.s.
43	** *HLADRB1* **	**5518**	**10,901**	**0.50**	**0.82**	**<0.0001**
44	** *IFIH1* **	220.4	281	-	n.s.	n.s.
45	** *IFNG* **	147	139	-	n.s.	n.s.
46	** *IGF2BP2* **	1118	1129	0.24	n.s.	n.s.
47	** *IL12B* **	69	56.3	-	n.s.	n.s.
48	** *IL13* **	188	117	1.60	n.s.	n.s.
49	** *IL18* **	**69**	**336**	**0.20**	**0.95**	**<0.0001**
50	** *IL1B* **	793	237	-	n.s.	n.s.
51	** *IL2* **	44	49	-	n.s.	n.s.
52	** *IL21* **	102	128	-	n.s.	n.s.
53	** *IL2RA* **	240	256.2	-	n.s.	n.s.
54	** *IL7RA* **	259	463	-	n.s.	n.s.
55	** *INS* **	12.2	12.7	-	n.s.	n.s.
56	** *IRF5* **	388	483	-	n.s.	n.s.
57	** *ITGAM* **	349	345	-	n.s.	n.s.
58	** *ITGAV* **	**893**	**2247**	**0.39**	**0.98**	**<0.0001**
59	** *ITPR3* **	1698	735	2.31	n.s.	n.s.
60	** *JAZF1* **	45	61	-	n.s.	n.s.
61	** *KIAA0350* **	627	473	-	n.s.	n.s.
62	** *KLC1* **	999	745	-	n.s.	n.s.
63	** *KLRB1* **	80	141	-	n.s.	n.s.
64	** *LAG3* **	74	83	-	n.s.	n.s.
65	** *MDR1* **	127	141	-	n.s.	n.s.
66	** *MERTK* **	228	235	-	n.s.	n.s.
67	** *MYO9B* **	**856**	**278**	**3.07**	**0.97**	**<0.0001**
68	** *NELL1* **	54	148	0.36	n.s.	n.s.
69	** *NOD2* **	**51**	**272**	**0.18**	**0.95**	**<0.0001**
70	** *NOTCH2* **	551	1075	-	n.s.	n.s.
71	** *NRAS* **	316	320	-	n.s.	n.s.
72	** *PADI4* **	54	59.7	-	n.s.	n.s.
73	** *PDE4B* **	412	349	-	n.s.	n.s.
74	** *PPARG* **	34	139	-	n.s.	n.s.
75	** *PTEN* **	1554	1172	-	n.s.	n.s.
76	** *PTPN2* **	756	1174	-	n.s.	n.s.
77	** *PTPN22* **	54.4	73	-	n.s.	n.s.
78	** *REL* **	**116.5**	**289**	**0.40**	**0.8**	**<0.0001**
79	** *RPL5* **	108	78	3.33	n.s.	n.s.
80	** *RSBN1* **	283	344	-	n.s.	n.s.
81	** *RUNX3* **	2620	849	3.08	n.s.	n.s.
82	** *SH2B3* **	616	417	-	n.s.	n.s.
83	** *SLC22A4* **	185	206	-	n.s.	n.s.
84	** *SNCA* **	**1758**	**683**	**2.57**	**0.92**	**<0.0001**
85	** *STAT2* **	419	496	-	n.s.	n.s.
86	** *STAT3* **	1406	1333	-	n.s.	n.s.
87	** *STAT4* **	303.6	389	-	n.s.	n.s.
88	** *THADA* **	626	596	-	n.s.	n.s.
89	** *TLR2* **	222	264	-	n.s.	n.s.
90	** *TNFAIP3* **	733	581	-	n.s.	n.s.
91	** *TNFSF11* **	67	100	-	n.s.	n.s.
92	** *TRAF1* **	399.13	342	-	n.s.	n.s.
93	** *TSHR* **	33.8	42.1	-	n.s.	n.s.
94	** *TYK2* **	**1126**	**725**	**1.55**	**0.86**	**<0.0001**
95	** *TYRO3* **	**375**	**859**	**0.43**	**0.81**	**<0.0001**
96	** *VEGFA* **	1121	622	1.80	n.s.	n.s.
97	** *WFS1* **	1097	974	-	n.s.	n.s.
98	** *XBP1* **	731	818	-	n.s.	n.s.

**Table 3 cancers-14-00991-t003:** The expression of the four validated genes was analyzed in all cancer types, in the GEPIA2 database. Asterisks indicate whether that specific cancer shows a significantly altered expression of that specific gene, compared to the corresponding healthy controls. Most of the significant differential expressions are found in cancers showing the higher “knowledge-based Autoimmunity Score” (kb-AS).

Cancer Type	Autoimmunity Score	NOD2	BAX	IL-18	ADRB2
Cervical	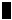				
Esophagus	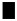	*		*	*
Pheoc-Para	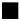				
Endomterial	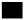			*	*
Rectum	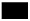				
Adrenocortical	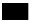			*	
Prostate					*
Mesothelioma					
Head&Neck	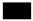				
Ovary	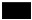				
Glioblastoma	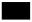			*	
Liver	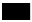				
Sarcoma	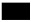				
Bladder	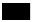				
Testis	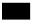				
Colon	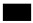			*	
Kidney	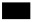				
Breast	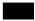				*
Lung	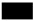				*
Glioma				*	
Gastric	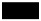			*	
Leukemia	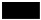	*		*	*
Pancreatic	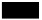	*		*	
Melanoma	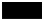	*	*	*	*
Cholangio	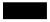		*	*	*
Thyroid	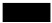		*	*	*
Lymphoma	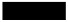	*	*	*	
Thymoma	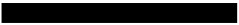	*	*	*	
	0 2 4 6 8 10 12				

## Data Availability

Raw data regarding expression values from GEO and from Ist Online are deposited in Mendeley data at https://data.mendeley.com/GEO. Data were from https://www.ncbi.nlm.nih.gov/sites/GDSbrowser?acc=GDS1375GEPIA2. Analysis was carried out on http://gepia2.cancer-pku.cn/ Ist Online data were from http://ist.medisapiens.com (currently not available). GENT2 analysis was carried out on http://gent2.appex.kr/gent2/.

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
