# Peer review of "Expression of Autoimmunity-Related Genes in Melanoma"

_cancers, 2022, doi:10.3390/cancers14040991_

Round 1

Reviewer 1 Report

Minor comments:

1. Figure 7

Comment: It needs to be revised. One of the axes must be with survival (months) and the other with expression level.

2. `The present study suggests for the first time that NOD2 expression levels may be indeed related to melanoma onset`.

Comment: Low or high level of NOD2 expression (to be specified)?  

3. `Patients showing high-expression of NOD2 and of IL-18 also show a significant survival improvement as compared to low-expression patients`.

Comment: Is it NOD2 involved in both the onset and evolution of melanoma? If so, please note that NOD2 is also involved in:

embryogenesis (https://pubmed.ncbi.nlm.nih.gov/25674246/),

regeneration (https://pubmed.ncbi.nlm.nih.gov/22951952/) and

adaptation (https://pubmed.ncbi.nlm.nih.gov/12478238/).

This means that the relationship between NOD2 and melanoma/ cancer should not be limited to the immune system, but also to other mechanisms related to carcinogenesis and the evolution of malignancy, as described by: https://pubmed.ncbi.nlm.nih.gov/34695580/  .

Author Response

Reviewer 1

We thank the Reviewer for the comments. Below please find the point-by point reply.

Question 1 : Figure 7

Comment: It needs to be revised. One of the axes must be with survival (months) and the other with expression level.

Answer: As requested, axes names were substituted with “Survival (months)” and  “expression level” .

Question 2: Comment: Low or high level of NOD2 expression (to be specified)? 

Answer: Lines 337-338 in the Discussion section state that: “In all four databases investigated, NOD2 expression appears strongly and significantly reduced in melanoma patients”.

Within the results, Table 2 reports at pag 7 the NOD2 reduction in GEO database; Figure 3 shows the NOD2 reduction in Ist Online database; Figure 4 shows the NOD2 reduction in GEPIA2 database; lines 211-212 report the NOD2 reduction in GENT2.

Question 3:  If so, please note that NOD2 is also involved in:

embryogenesis (https://pubmed.ncbi.nlm.nih.gov/25674246/),

regeneration (https://pubmed.ncbi.nlm.nih.gov/22951952/) and

adaptation (https://pubmed.ncbi.nlm.nih.gov/12478238/).

This means that the relationship between NOD2 and melanoma/cancer should not be limited to the immune system, but also to other mechanisms related to carcinogenesis and the evolution of malignancy, as described by: https://pubmed.ncbi.nlm.nih.gov/34695580/  .

Answer: We thank the Reviewer for this note. The following sentence was added in the Discussion section at lines 341-345, according to the Reviewers’ suggestions.

 “NOD2 is involved in many processes such as embryogenesis (41) and in regeneration (42) and chronic inflammation (43), suggesting that the relationship between NOD2 and melanoma/cancer may not be limited to the immune system, but may involve also other mechanisms related to carcinogenesis and the evolution of malignancy (44)”.

Reviewer 2 Report

In this manuscript, Scatozza and colleagues conduct a systems biology-based study to find links between melanoma and autoimmunity. Although the idea is interesting and has a potential to shed light on autoimmune diseases that frequently co-occur with melanoma, there are many concerns regarding the robustness of the study design, which reduce the scientific merit of the study.

The most important concern is the reference list of autoimmunity-related genes which the authors chose and subsequently juxtaposed with differentially expressed genes (DEGs) in melanomas. This list of 98 genes derives from 1) one Hindawi publication summarizing genes predisposing to several autoimmune diseases (ADs) and 2) one popular science website (eupedia). This selection severely biases their analysis, because on one hand, it is not clear whether the data from eupedia have undergone any kind of peer-review process; and on the other hand, predisposing factors of AD are not directly comparable to genes deregulated in a tumor lesion. It is also not clear which aspects of the autoimmunity reaction are affected by deregulation of these 98 genes. Are these genes deregulated in the immune-reactive tissue or in the immune cells that attack the affected tissue?

Compiling a list of autoimmunity-related genes was the most crucial step of the pipeline, but the authors seem to neither have taken into account the abovementioned aspects, nor have compiled it in a comprehensive, organized and reproducible manner. This approach eventually reduces the strength of their subsequent metaanalyses. The fact that the reference gene list is rather small (containing only 98 genes) and only a 6% overlap was found with the DEGs in melanomas also renders the data non-convincing.

The second major concern is the lack of any kind of positive or negative controls. Out of the 98 genes, the authors find 6 genes deregulated in melanomas, which they subject to validations in several datasets from melanoma tumors versus normal controls. This comparison introduces another major bias: first of all, although the number of patient samples is mentioned, the number of DEGs in melanomas is not mentioned. The 98 genes were compared against how many DEGs? When one compares such a small reference list of 98 genes with another gene list (especially a much larger one), some overlapping between the two lists is expected, due to either stochasticity events or the pleiotropic nature of the common genes. The mere fact that 6 autoimmunity-related genes that are also deregulated in melanomas does not essentially imply autoimmunity features of melanomas. To demonstrate that these commonalities are not due to stochastic event, the authors should have used a random list of equal number of autoimmunity-related genes as a negative control, which they should also compare versus the DEGs in melanomas. If the percentage of autoimmunity-related genes that are deregulated in melanomas remains significantly higher than the random list genes that deregulated in melanomas, this suggests deterministic trend.

Melanoma is essentially a highly immunogenic lesions which provokes immune responses via the expression of melanocytic-differentiation antigens. Since these immunogenicity markers are widely known and are sometimes related with spontaneous disease regression, they could have been used as positive controls: the expression of autoimmunity-related genes could have been correlated with the expression of these markers, to link the autoimmunity-related genes with immunogenicity.

Overall, this is an interesting idea, but the experimental design lacks robustness and fails to exclude pseudocorrelations in a definite manner.

Author Response

Reviewer 2

Question n. 1: “This selection severely biases their analysis, because on one hand, it is not clear whether the data from eupedia have undergone any kind of peer-review process”

Answer:   Thanks to the Reviewer for this accurate criticism. As correctly highlighted by Reviewer n. 2, the list of 98 autoimmunity-related genes is indeed not exhaustive. The sentence at line 313-316 within the Discussion section has been modified to underline this point.

The most complete list of autoimmune-diseases-related genes is reported in the GAAD database (https://pubmed.ncbi.nlm.nih.gov/30268934/) available at the link http://gaad.medgenius.info/genes/. This database contains 4186 genes. Due to the lack of proper informatic tools, we decided to select and validate only 98 genes, taken from this list and also reported in References 17 and 18. To comply with the reviewer’s comment, we add the citation of the GAAD reference and substituted Ceccarelli et al reference with the more appropriate Gregersen et al. reference  (see lines 313-316).

Therefore, all genes reported in the 98-genes list have referenced association with autoimmune diseases according to the references 17, 18, 19 of the revised manuscript.

The exhaustive analysis on all 4186 genes is currently going on with a bioinformatic tool under development.

The Methods section has been modified at line 411.

Question n. 2:  “….and on the other hand, predisposing factors of AD are not directly comparable to genes deregulated in a tumor lesion.”

Answer:  We thank the Reviewer for this comment. The gene expression of AD-related genes in melanoma samples is in fact never been investigated before and is the main object of the current study.

Question n. 3: “…It is also not clear which aspects of the autoimmunity reaction are affected by deregulation of these 98 genes. Are these genes deregulated in the immune-reactive tissue or in the immune cells that attack the affected tissue?”

Answer: we focused our analysis on autoimmune-related genes, postponing to future studies addressing specific functional relationships.

Question n. 4: “The fact that the reference gene list is rather small (containing only 98 genes) and only a 6% overlap was found with the DEGs in melanomas also renders the data non-convincing.”

Answer: The relatively small number of differences (6 out of 98 genes) is due to the severe thresholds selected (fold change < 0.5 or >1.5; AUC > 0.93 p < 0.0001).

In fact, more genes (i. e.12 out of 98) show a significant alteration, using less severe thresholds (fold change < 0.5 or >1.5; AUC > 0.8;  p < 0.0001), as clearly reported in Table 2.

Question n. 5: “The second major concern is the lack of any kind of positive or negative controls.”

Answer:  thanks for this comment. It is indeed an interesting issue: A proper negative control would be a gene known to be NOT involved in autoimmunity. This information is rather difficult to find. In fact, data reported in the literature only refer to genes found to be involved in autoimmunity (i.e., positive controls) while genes not-involved are not indicated. This makes difficult the possibility to identify proper negative controls.

Further, the required positive controls are the 98 genes selected for the preliminary analysis.

Question n. 6: “Although the number of patient samples is mentioned, the number of DEGs in melanomas is not mentioned.”

Answer: In the study design followed in the current manuscript, we analyze in each validation step only those genes identified in the previous step, as clearly indicated in the flow depicted in Figure 1. Therefore, in validation step n. 1, 6 genes were investigated since those 6 were selected in the previous screening step; in validation step n. 2, 6 genes were investigated since those 6 were validated in the previous step;

Finally, in validation step n. 3, 4 genes were investigated since those 4 were validated in the previous step.

Question n. 7: “Since these immunogenicity markers are widely known and are sometimes related with spontaneous disease regression, they could have been used as positive controls: the expression of autoimmunity-related genes could have been correlated with the expression of these markers, to link the autoimmunity-related genes with immunogenicity.”

Answer: Correlating autoimmunity to expression of regression markers is a very interesting point and we thank the Reviewer for this suggestion.

Reviewer 3 Report

The manuscript "Expression of autoimmunity-related genes in melanoma" describe the validation of 4 genes related to autoimmunity in malignant melanomas, through a validation using 4 different publicly accessible databases.

The article is well written and the study design is sound.

Author Response

Reviewer n. 3

We thank Reviewer n.3 for the positive evaluation.